# Tumor-Associated Edema in Children with Kaposi Sarcoma: 14 Years’ Experience at Kamuzu Central Hospital, Lilongwe, Malawi

**DOI:** 10.3390/cancers16223769

**Published:** 2024-11-08

**Authors:** Fatsani Rose Manase, Allison Silverstein, William Kamiyango, Jimmy Villiera, Clement Dziwe, Claudia Wallrauch, Tom Heller, Mark Zobeck, Tamiwe Tomoka, Michael E. Scheurer, Carl E. Allen, Nmazuo Ozuah, Rizine Mzikamanda, Nader Kim El-Mallawany, Casey L. McAtee

**Affiliations:** 1Baylor College of Medicine Children’s Foundation, Lilongwe P.O. Box B-397, Malawi; fmanase@baylor-malawi.org (F.R.M.); wkamiyango@baylor-malawi.org (W.K.); rmzikamanda@baylor-malawi.org (R.M.); 2Texas Children’s Hospital Global HOPE, Lilongwe P.O. Box B-397, Malawi; jvilliera@unclilongwe.org (J.V.); ceallen@texaschildrens.org (C.E.A.); nmazuo.ozuah@bcm.edu (N.O.); 3Department of Pediatrics-Palliative Care Medicine, University of Colorado School of Medicine, Aurora, CO 80045, USA; allison.silverstein@childrenscolorado.org; 4University of North Carolina Chapel Hill Project Malawi, Lilongwe P.O. Box A-104, Malawi; ttomoka@unclilongwe.org; 5Lighthouse Clinic, Lilongwe P.O. Box 106, Malawi; cleedziwe@gmail.com (C.D.); cwallrauch@lighthouse.org.mw (C.W.); theller@lighthouse.org.mw (T.H.); 6Institute of Infectious Disease and Tropical Medicine, LMU University Hospital Munich, 80802 München, Germany; 7International Training and Education Center for Health, University of Washington, Seattle, WA 98195, USA; 8Baylor College of Medicine, Houston, TX 77030, USA; mark.zobeck@bcm.edu (M.Z.); scheurer@bcm.edu (M.E.S.); nader.el-mallawany@bcm.edu (N.K.E.-M.); 9Kamuzu Central Hospital, Lilongwe P.O. Box 30377, Malawi

**Keywords:** Kaposi sarcoma, Africa, LMIC, HIV-related malignancies, pediatric oncology

## Abstract

This research is focused on Kaposi sarcoma (KS), a cancer occurring usually among children and adults living with HIV. Despite improvements in HIV treatments, KS remains common in eastern and central Africa. KS in children manifests in many different ways, from small skin lesions to cancer throughout the body. KS edema is a version of the disease where patients develop hard, wood-like skin lesions causing decreased mobility and quality of life. This study analyzes data from over a decade of treating pediatric KS in Malawi and aims to better describe the disease in children. The findings emphasize the chronic nature of KS edema, its propensity to relapse, its impact on survival, and the need for improved long-term management strategies to reduce relapses and progression. This research provides important insights into the unique biology and clinical presentation of KS in children, helping to guide future treatment approaches in resource-limited settings.

## 1. Introduction

Kaposi sarcoma (KS) is an angioproliferative lymphoid tumor driven by infection with human herpesvirus-8 (HHV-8) [1,2,3]. KS is extraordinarily heterogenous, variably affecting the skin, mouth, lymph nodes, and internal organs [1,4,5]. Of its four epidemiological subtypes, the disease in children most often occurs in its epidemic, HIV-related form; less commonly, pediatric KS occurs in its HIV-unrelated, endemic form unique to children living in central and eastern Africa [6].

In contrast to adult KS where prototypical hyperpigmented mucocutaneous nodules and plaques are its most common manifestation, children present most commonly with bulky lymphadenopathy and cytopenias with or without characteristic skin lesions [3,7,8]. KS without cutaneous disease is relatively common in children, occurring in approximately half of patients, compared with <5% in adults [9,10].

The Lilongwe Staging Classification for KS was developed to capture the clinical heterogeneity of the disease in children, assigning stages to its cutaneous, lymphadenopathic, edematous, and visceral/disseminated presentations that are not well captured in the established TIS system used to classify HIV-related KS [1,9]. While most of the pediatric KS literature has focused on its more common lymphadenopathic and visceral presentations, little research has focused on defining the KS edema subtype.

KS tumor-associated edema, often referred to as “woody edema” because of its firm, wood-like cutaneous skin lesions, typically affects the lower extremities and occurs in about a third of KS cases [3,11,12,13]. Clinically, it is identified by hard, non-pitting edematous skin, often accompanied by hyperpigmented, ulcerated, or fungating lesions (Figure 1). These complex lesions consist of tumor cells, dysplastic lymphatic hyperplasia, lymphostasis, and inflammatory infiltrates [14,15]. Though KS edema is distinct from other forms of lymphedema, various terms such as “KS tumor-associated edema”, “woody edema”, and “KS lymphedema” are used to describe these neoplastic lesions [1,14,16].

KS edema is an understudied but important KS subtype due to its tendency to cause chronic, debilitating disease in children [1,3]. It often severely impacts children’s quality of life by restricting normal play, limiting ambulation, and contributing to social stigma. There is a paucity of research in children focusing on the biology, clinical progression, and outcomes of KS edema.

The purpose of this study is to better define KS edema as a unique clinical phenotype of pediatric KS. Here, we leveraged over a decade of experience treating pediatric KS in Malawi to conduct this first-of-its-kind clinical study focused specifically on KS edema in children. We show that KS edema is a distinct clinical phenotype of pediatric KS and that, while it carries a higher short-term survival rate relative to other phenotypes, children with KS edema remain at high risk of long-term mortality due to a uniquely high risk of relapse and progression.

## 2. Materials and Methods

### 2.1. Study Design

The primary objective of this study was to characterize KS edema in children in support of the hypothesis that pediatric KS presents with distinct clinical subtypes. To achieve this, we conducted a retrospective cohort study of children diagnosed with KS at Kamuzu Central Hospital in Lilongwe, Malawi, between 2010 and 2023. This study was a sub-cohort analysis of children with KS edema at either initial diagnosis or relapse. Data describing the larger cohort in aggregate have been published previously [17].

Data were manually extracted from electronic records. STROBE reporting guidelines were followed [18]. Data were complete with exceptions noted in the results. Ethical approval was obtained in Malawi and the United States. Signed consent was obtained from the guardian of the patient with specific permissions to publish the photograph in Figure 1.

### 2.2. Diagnosis

KS edema defines Lilongwe Stage 3 within the Lilongwe Staging Classification System for pediatric KS [1,19]. Stage 3 includes children with KS edema with or without features of Stages 1 (limited skin/oral) and Stage 2 (lymph node-predominant) disease. KS edema can occur in isolation, and when features of Stage 1 and/or Stage 2 disease are present, they are typically less prominent in children with concurrent KS edema; therefore, any patient with KS edema who does not have visceral or disseminated cutaneous disease is classified as Stage 3 in the Lilongwe system [1,3]. Children with visceral (i.e., pulmonary, gastrointestinal) or disseminated cutaneous disease are classified as Stage 4, regardless of the presence of edema. Children with Stage 4 disease plus KS edema are included in this study (17% of cohort) but analyzed separately where noted. Detailed diagnostic criteria for visceral KS in Malawi where endoscopy/bronchoscopy are not available are reviewed elsewhere [20]. Briefly, visceral disease is most often diagnosed clinically based on severe pulmonary or gastrointestinal disease in patients with known KS not otherwise explained by other etiologies (e.g., infection).

Biopsies were routinely obtained of KS lesions, but some cases of KS edema were diagnosed clinically due to the difficult anatomical location of the biopsy (e.g., in the groin) and the presence of pathognomonic clinical features (e.g., hyperpigmented skin lesions) to guide diagnosis. When biopsies were obtained, they were stained with hematoxylin and eosin for morphology and HHV-8 Latency Nuclear Antigen immunostaining for confirmation [21].

### 2.3. Treatment and Outcomes

The treatment of KS at the center evolved over the study period based on emerging data supporting the use of paclitaxel for upfront therapy [1,22,23]. The current treatment paradigm at our center is provided in Table 1. The specific therapy received by patients is reviewed in the results.

Response status was reported at each clinic follow-up visit by healthcare staff. The response categories were not applied with strict adherence to a prospective research protocol, but clinic staff received specific training to code responses according to the following statuses: (1) complete response (CR): total disappearance of all KS features on physical exam; (2) partial response (PR): children who did not achieve CR but achieved clinical improvement that remained stable, without progression after completing chemotherapy; (3) progressive disease: disease progressing in patients who had not achieved CR (e.g., progressing after having achieved stable disease); (4) relapse: progressive disease in patients previously in CR [10]. A subtle, user-dependent overlap and imprecision in outcome coding likely occurred over the study period. We report levels of response here to highlight the spectrum of responses seen in KS edema, from a virtual disappearance of the disease to a minimal change with stable lesions off chemotherapy.

All patients were assessed for progressive disease or relapse at routine follow-up appointments. Patients were coded in the medical record as CR, PR, progressive disease, or relapse based on history and physical exam at each follow-up appointment. Disease status was determined by the attending clinician at each clinic visit.

Event-free survival was calculated as the time from diagnosis to (1) failure to achieve documented CR after the initial course of chemotherapy; (2) progressive disease in patients with PR; (3) relapse in patients with CR; or (4) death, whichever is earliest. Progression-free survival was calculated among patients in CR or PR off of chemotherapy between diagnosis and progression, relapse, or death. Overall survival was the time from diagnosis to death. Patients lost to follow-up were censored. Treatment abandonment was not recorded. All surviving patients were right-censored on 31 December 2023.

### 2.4. Statistical Analysis

Continuous variables were compared using Welch’s *t*-test or the Wilcoxon rank-sum test, and categorical variables were compared with Pearson’s or Fisher’s χ^2^ test. Relative risk was used to quantify risk effect sizes. Survival was calculated using the Kaplan–Meier method. Median follow-up time was calculated using the reverse Kaplan–Meier method [24]. Group survival comparisons were made using the log-rank test. All hypothesis tests were two-sided with α = 0.05. All analyses were performed in R (version 4.3.3) with special use of the Survival and EpiR packages [25,26,27].

## 3. Results

### 3.1. Patient Characteristics at Diagnosis

We identified 193 patients with KS, among whom 52 (27%) had KS edema at diagnosis, comprising the cohort of focus in this study. Within this cohort, 39 (75%) had biopsy-confirmed lesions. The median follow-up time within the cohort was 7 years (interquartile range, IQR, 5–9 years).

Characteristics of the cohort are provided in Table 2. Among those children with KS edema, nine (17%) presented with visceral (pulmonary or gastrointestinal) disease and/or disseminated skin/oral disease (Lilongwe Stage 4). KS edema was most common in the lower extremities (90% of cases).

A majority of patients with HIV (23/42, 55%) were on antiretroviral therapy at the time of KS diagnosis. Viral load was available at diagnosis for only 15/42 patients with HIV-related KS. Among these patients, the median HIV viral load was 112,896 (IQR, 275,000–202,503) and 4/15 (27%) had an undetectable viral load at diagnosis.

When compared to the larger KS cohort without KS edema, children with edema were older at a median age of 12 years (IQR, 9–13 years) versus 6 years (IQR, 4–11; *p* < 0.001). Thrombocytopenia (platelets < 150,000/µL) was less common among patients with edema compared with those without it (28% versus 51%, *p* = 0.01).

### 3.2. Therapy, Outcomes, and Relapse

Three patients (7% of HIV-related KS cohort) were treated initially with antiretroviral therapy alone. The remainder were started on systemic chemotherapy. Among patients with Stage 3 disease, the majority (88%) received bleomycin and vincristine for eight cycles as frontline therapy. Among patients with Stage 4 disease, most (67%) received bleomycin and vincristine plus doxorubicin for a median of 10 cycles (IQR, 8–10).

Following first-line therapy, 48/52 patients (92%) demonstrated a clinical response, with most patients (48%) achieving partial response and able to discontinue chemotherapy. The remainder of those responding (44%) achieved documented CR. Four patients (8%) progressed on therapy, two of whom presented initially with Stage 4 disease.

The patterns of partial response were heterogenous among Stage 4 patients. In two patients with disseminated cutaneous disease plus KS edema, the skin disease improved to Stage 1 disease with a complete resolution of edema. Two patients with Stage 4 disease (one with pulmonary disease, the other with disseminated cutaneous disease) had a resolution of the Stage 4 disease components but had residual KS edema, shifting them to Stage 3 classification. Finally, a single patient experienced a resolution of KS edema but had partially responsive pulmonary disease.

A pattern of relapse and progression following remission and partial response was common (Figure 2). Relapse following complete response or progression following partial response occurred in 24 (46%) patients. Relapse and progressive disease occurred at the site of old KS edema lesions in most cases. The median time to first relapse/progression was 12 months from diagnosis (IQR, 8–18). Sixteen patients (31%) had two or more relapses or progressions. The median time between all relapses or progressions was 13 months (IQR, 10–16). There was no difference in the median time to relapse/progression between those progressing/relapsing as Lilongwe Stage 3 versus Stage 4.

Therapy for relapsed or progressive disease evolved over time and was generally tailored for individual patients (Table 3). A subsequent regimen of bleomycin–vincristine was the most common second-line therapy for relapsed/progressive KS edema. Over time, paclitaxel was used more frequently for subsequent relapses as it became more available in Malawi, and evidence of its safety and efficacy in low-resource settings increased.

Among those patients experiencing relapse or progressive disease, patients had a median of three relapse/progression events (IQR, 1–3), ranging up to eight events in a single patient and occurring up to ten years after initial diagnosis in a child with Stage 3 endemic KS. Of 66 relapse/progression events in the cohort, most (74%) were relapses or progressions with KS edema (Stage 3). Eight patients (12%) relapsed with Stage 4 disease, five of whom originally presented with Stage 3 disease, three with endemic KS, and two with HIV-related KS (Table 4). Notably, among the larger cohort of patients with KS initially presenting without KS edema (i.e., patients with Stage 1, Stage 2, and Stage 4 without KS edema), zero patients relapsed with KS edema.

Four of nine patients presenting initially with Stage 4 disease experienced relapse or progression, one with one or more relapses/progressions as Stage 3 disease, and three as Stage 4 again alongside KS edema. A single patient originally diagnosed with Stage 4 disease had three recurrences of KS edema following the resolution of visceral disease, before ultimately dying with relapse as Stage 4 disease 5.5 years following their initial diagnosis. All patients who relapsed with Stage 4 disease relapsed with pulmonary KS.

Children with endemic KS relapsed more often than those with HIV-related KS (70% vs. 41%), but this likely underpowered analysis did not reach statistical significance (*p* = 0.2). Among those children with relapses or progressions, children with endemic KS had a higher number of relapse/progression events (median = 3, IQR, 3–3) compared with children with HIV-related KS (median = 2, IQR 1–3; *p* = 0.04).

Leveraging the larger cohort of all KS patients (all Lilongwe Stages, n = 193), we assessed whether relapse or progression was more likely among patients initially presenting with KS edema (n = 52) versus those without it (n = 141). Patients initially presenting with KS edema had a higher relative risk (RR) of one or more relapse/progression (RR = 2.1; 95%CI, 1.4–3.2; *p* < 0.001). Among those patients with KS who had one or more relapses/progressions, patients presenting initially with KS edema had a higher median number of total relapse/progressions from treatment response off chemotherapy: 3 (IQR, 1–3) versus 1 (IQR, 1–1.5; *p* < 0.001).

Event-free survival for patients with KS edema was 32% (95%CI, 22–48%) at two years and 24% (95%CI, 14–39%) at five years. Progression-free survival was 40% (95%CI, 29–56%) at two years and 31% (95%CI, 20–47%) at five years. Overall survival was 73% (95%CI, 62–86%) at two years and 57% (95%CI, 44–73%) at five years (Figure 3).

Survival among patients with KS edema stratified by initial presenting stage is shown in Table 5. Patients presenting initially with Stage 4 had a significantly worse progression-free survival (*p* = 0.009) and overall survival (*p* = 0.02) than those patients presenting with Stage 3 disease. There was no significant difference in event-free, progression-free, or overall survival between those with and without HIV infection (Table 5). Of note, among 35 children with HIV-related Stage 3 KS, overall survival at 5 years was 63% (95%CI, 48–82%). Among children with endemic KS presenting initially with Stage 3 disease, 5-year overall survival was 55% (95%CI, 27–100%).

Long-term outcomes of KS patients in the larger cohort of patients with KS are reported elsewhere [17]. For reference, 5-year overall survivals for patients with Lilongwe Stages 1, 2, and 4 were 82% (95%CI, 69–98%), 67% (95%CI, 56–79%), and 31% (95%CI, 19–50%), respectively.

Among patients in the cohort who died, 9 (39%) died of KS disease progression, 8 (35%) of complications related to HIV (e.g., infection), 3 (13%) of infection while on chemotherapy, and 3 (13%) of an unknown cause. Among children with HIV-related KS, mortality was most often due to HIV-related mortality rather than KS disease progression (44% vs. 22%). All patients with KS edema who died due to progression of KS died of visceral disease or lymphadenopathic (Stage 2) disease rather than due to direct complications of KS edema itself.

## 4. Discussion

### 4.1. Kaposi Sarcoma Edema: A Distinct Clinical Subtype

Kaposi sarcoma’s more subtle manifestation as tumor-associated edematous lesions is often overshadowed by the disease’s more dramatic presentations. This study leverages fourteen years’ experience treating KS in Malawi to highlight this unique disease subtype in children, adding to a growing body of evidence that KS edema is a distinct clinical phenotype of the cancer.

Although KS edema may advance slowly and is rarely fatal on its own, patients nonetheless face a substantial risk of recurrent disease and long-term mortality, potentially as late relapse with visceral disease. As discussed below, this disease pattern may provide clues to the underlying pathobiology of KS in the setting of edema.

Discussing KS edema as a distinct clinical phenotype in children arises from the risk-adapted Lilongwe Staging Classification for pediatric KS. The staging system, introduced in 2017, describes patients with overlapping yet distinct clinical phenotypes and outcomes [19]. In the Lilongwe system (named after Malawi’s capital and river of the same name), KS edema defines Stage 3 disease. In the adult staging system, KS edema is designated as T1 disease, grouped alongside visceral disease of the lungs and intestines [28].

Similarly to previous reports, we show that children with visceral disease (Stage 4) with or without edema have significantly lower progression-free and overall survival than those with edema alone. The stark differences in outcomes between these two phenotypes historically grouped together within the T1 staging group have also been recently discussed in the adult literature [28].

For these reasons, children with visceral or disseminated disease, whether or not they have lymph node involvement or edema, are classified as Lilongwe Stage 4. At our center, these patients have traditionally received an intensified chemotherapy regimen with an anthracycline (e.g., doxorubicin) added to bleomycin and vincristine, although recent adult data showing the effectiveness and safety of paclitaxel have shifted our approach to using frontline paclitaxel when available due to a favorable side-effect profile and the ability to avoid the chronic toxicity of bleomycin- and anthracycline-containing regimens [22,23].

The long-term relapsing and remitting pattern of KS edema, cycling over many years between stable responses off chemotherapy and onwards again to progressive disease, distinguishes it from other forms of KS. Children with KS edema are more likely to have a return of their disease than other KS subtypes, and when they have relapses or progressions, they occur more often. KS edema often requires repeated cycles of chemotherapy spanning multiple years to maintain disease control over time. For example, a single patient experienced eight progressions over eight years, requiring cytotoxic chemotherapy each time to achieve sustained clinical stability and a return of mobility. It is therefore essential that patients with KS edema be monitored long-term for a potential return of their disease.

### 4.2. KS Edema and Mortality

While previous reports show that the KS edema subtype has a higher short-term survival than its lymphadenopathic and visceral or disseminated counterparts, long-term all-cause mortality in the cohort was high [10,20]. Patients presenting initially with KS edema without visceral or disseminated disease typically have a high two-year overall survival ≥80% [10,19]. In this cohort with an extended follow-up (median seven years), overall survival in the edema cohort fell to 57% at five years. Among patients with KS edema without evidence of visceral or disseminated disease at diagnosis, 5-year overall survival was only marginally higher at 61%.

Among children with HIV-related KS, excess mortality was often related to complications of HIV (e.g., pneumonia) rather than KS progression itself (44% versus 22% of deaths). In the modern antiretroviral era, 5-year survival among children living with HIV in Malawi and neighboring countries is expected to approach 90% [29,30,31,32]. Among 35 children in this cohort with HIV-related Stage 3 KS, survival at 5 years was only 63%, highlighting these children living with HIV as a subgroup with an extraordinarily high all-cause risk of death with a clear indication for close surveillance.

Low long-term survival was not limited to those children living with HIV. Children with endemic, HIV-negative KS presenting initially with Stage 3 disease had a 5-year overall survival of only 55%, indicating that a sustained risk of mortality is present even among children without the underlying mortality risks associated with HIV. An illustrative case is of a boy in the cohort dying of pulmonary KS nearly seven years after presenting initially with Stage 3 disease. A second child with initial Stage 3 disease died of a final relapse as pulmonary KS following two relapses as KS edema with a near-complete resolution of the disease and stability off chemotherapy between relapses.

As seen in these examples, long-term mortality in children with KS edema was often related to the ability of the disease to return as visceral disease, often many years after the initial diagnosis as edema alone. Edema itself, though often causing lost ambulation, chronic pain, and social stigma, was not directly responsible for any deaths. KS disease-related deaths were in every patient due to progression to fulminate lymphadenopathic or visceral disease. Treatment-related mortality due to febrile neutropenia, although rare, occurred in three patients, underscoring the risks of indefinite therapy for patients requiring chemotherapy for ongoing recurrences.

### 4.3. Potential Mechanisms of Relapse and Progression

Although the mechanism underlying repeated occurrences of KS edema and the potential progression to fulminant disease is not fully understood, it may be related to unique characteristics of the tumor microenvironment in chronically edematous tissue. All forms of KS are driven by the dysregulated cellular proliferation, angiogenesis, inflammation, and immune escape caused by HHV-8 infection, typically but not always in the setting of an underlying immune deficiency [2,13].

When biopsied, KS edema lesions show tumor aggregates of the pathognomonic spindle cell of KS, likely a malignantly transformed HHV-8-infected lymphatic endothelial or mesenchymal precursor cell [15,21,33]. In KS edema lesions, KS spindle cells are surrounded by dysplastic and disorganized lymphatic vessels [15]. These thin-walled, dilated, and dysfunctional vessels become occluded by endothelial hyperplasia, lymphadenopathy, and/or KS spindle cells themselves, leading to local edema and fibrosis [14].

The lymphedematous tissues provide a pro-neoplastic environment through many potential mechanisms. Edematous tissue exhibits diminished immune surveillance and recruitment of T cells, antigen-presenting cells, and macrophages, creating a privileged tumor microenvironment for HHV-8-driven cellular proliferation and immune escape [34,35]. Lymphedematous tissue further provides an environment where HHV-8 viral proteins may persist in interstitial tissue, promoting inflammation, angiogenesis, and neoplasia [14].

That new KS edema lesions most often recur at sites of old edema lesions suggests that local edema and fibrosis may create a reservoir for future relapses and progressions in the setting of lifelong HHV-8 infection within the microenvironment, even in KS lesions which may appear macroscopically resolved after therapy. Recurrences of KS edema appear to require cytotoxic chemotherapy for resolution in all cases, indicating the role that renewed neoplastic growth plays in relapsing/progressing cycles rather than that of chronic fibrosis and lymphatic damage alone.

Whether KS edema lesions themselves predispose children for relapses as visceral disease is unclear. Five patients in this cohort who presented originally with KS edema without visceral/disseminated disease relapsed with Stage 4 KS, with all five dying of their disease. Relapse with distant metastases following the initial presentation with local disease is common in other pediatric sarcomas, but to date, no study has matched the genetic clone of a localized de novo KS lesion to a distant relapsed, potentially metastatic lesion.

Robust research into the clonality of KS lesions is sparse, but adult investigations describe monoclonal cell populations alongside polyclonal populations within lesions, suggesting a theoretical potential for metastatic hematogenous spread independent of a local microenvironment maintained by HHV-8 infection [36]. The potential mechanisms of progression from low-stage to high-stage KS are underexplored and may provide a rich area for future research.

Alternatively, it is possible that some children harbor subclinical visceral lesions at initial presentation that seed future late-stage relapses. Adult series describe that patients with HIV-related KS may have subclinical visceral disease that is discovered only upon autopsy [37]. While children with visceral KS often have highly aggressive, fatal disease at presentation, lacking endoscopy and bronchoscopy in the resource-limited settings where pediatric KS occurs prohibits full staging workups to rule out mild or subclinical visceral disease that may have gone undetected in this study.

Therapy of KS edema in children remains limited mostly to cytotoxic chemotherapy. Complete and partial responses using the same medications used initially in subsequent cycles were often achieved, but failure to achieve a new response, especially in the presence of Stage 4 disease, was common. Patients in this cohort responded to traditional regimens of vincristine and bleomycin +/− anthracycline, as well as relatively newer monotherapy with paclitaxel [12,38]. Paclitaxel is superior to bleomycin and vincristine in adults, and it is safe and effective in the low-resource setting of Malawi [22,23]. Future clinical trials should include pediatric patients to replicate these findings in this understudied population of patients with KS.

Antiretroviral therapy alone, while often successful in treating some adult patients with KS, is almost always ineffective in children with KS [1]. Radiation, an older but effective therapy for KS, has been abandoned in children due to the risk of late effects and its general unavailability in low-resource settings [8]. Physical therapies such as compression therapy are only marginally effective in treating KS edema [39].

### 4.4. Strengths and Limitations

This study, the largest cohort study to date of patients with KS edema either in adults or children, has several strengths. This study leveraged 14 years of electronic records maintained by the Baylor College of Medicine International Pediatric AIDS Initiative (BIPAI) network. The KS clinical program at Kamuzu Central Hospital has pioneered the standardized approach to pediatric KS for over 20 years. Within this context, a comprehensive clinical and translational research program has created the environment for high-quality, robust research in a severely resource-limited setting.

This study is limited by its retrospective nature. While outcomes were well documented in the medical record, a prospective study with a standardized case report form would have better described the nuances of KS disease outcomes (e.g., partial response versus stable disease). Furthermore, the dispersed, resource-limited healthcare system of Malawi creates the possibility that some relapses and progressions were undiagnosed; however, the prolonged follow-up time within the cohort (median seven years) in relation to a median time to relapse of 12 months suggests that most if not all relapses and progressions were detected.

This study is also limited by a lack of biopsy-proven visceral lesions. While a general biopsy rate of 75% in the cohort is extraordinarily high in this resource-limited region, pediatric bronchoscopy and endoscopy are not available at the center; thus, visceral disease was diagnosed based on characteristic imaging, signs (e.g., GI bleeding), and typical KS lesions occurring alongside suspected visceral disease having failed therapy for non-KS etiologies.

## 5. Conclusions

KS edema is a clinically important and distinct KS subtype in children, characterized by a high risk for chronically relapsing and remitting disease, potential relapse as advanced-stage disease, and poor long-term survival. This study highlights the potential for progression to fatal visceral disease following many years of an indolent course with chronic edema. Further study is required to identify which patients are at highest risk for such progressions, especially considering its associated high rate of mortality. Furthermore, novel treatment approaches are necessary to prevent frequent relapses. Understanding the biology of pediatric KS remains extremely limited in comparison to pediatric cancers more common in high-income countries. Further funding and study of KS, like all pediatric cancers in resource-limited settings, is desperately needed to reduce the tremendous disparities in cancer outcomes between the wealthiest and poorest countries in the world.

## Figures and Tables

**Figure 1 cancers-16-03769-f001:**
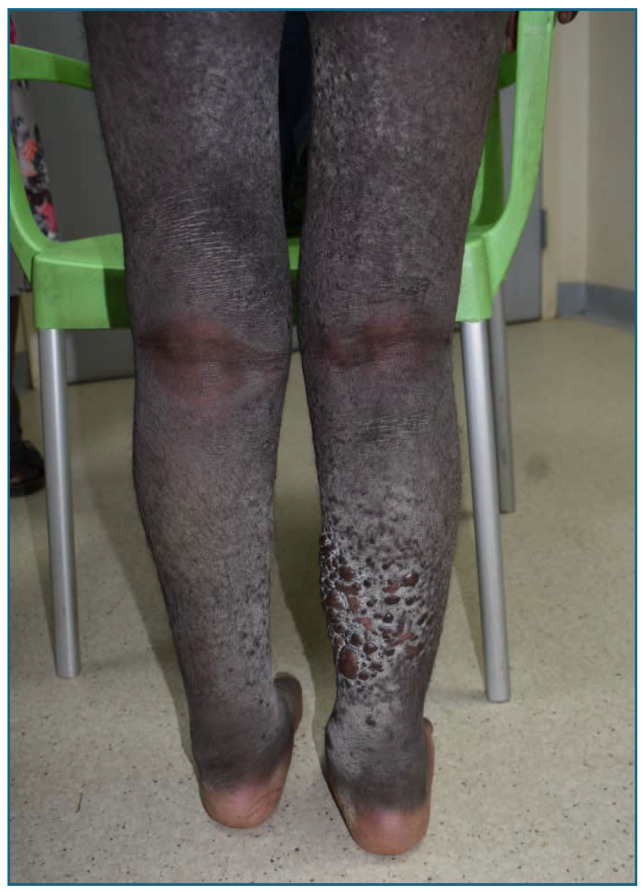
Kaposi sarcoma edema in the lower extremities. Kaposi sarcoma edema is observed extending from the upper thigh inferiorly to the ankles with overlying hyperpigmentation and firm bullae.

**Figure 2 cancers-16-03769-f002:**
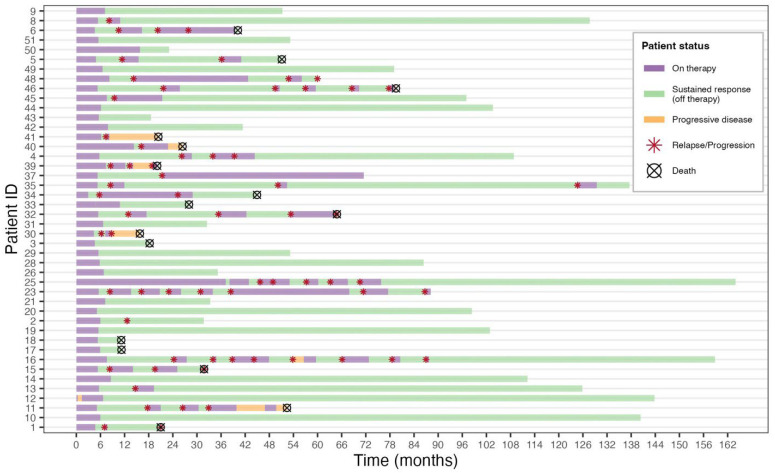
Swimmer plot of outcomes of patients with Kaposi sarcoma edema.

**Figure 3 cancers-16-03769-f003:**
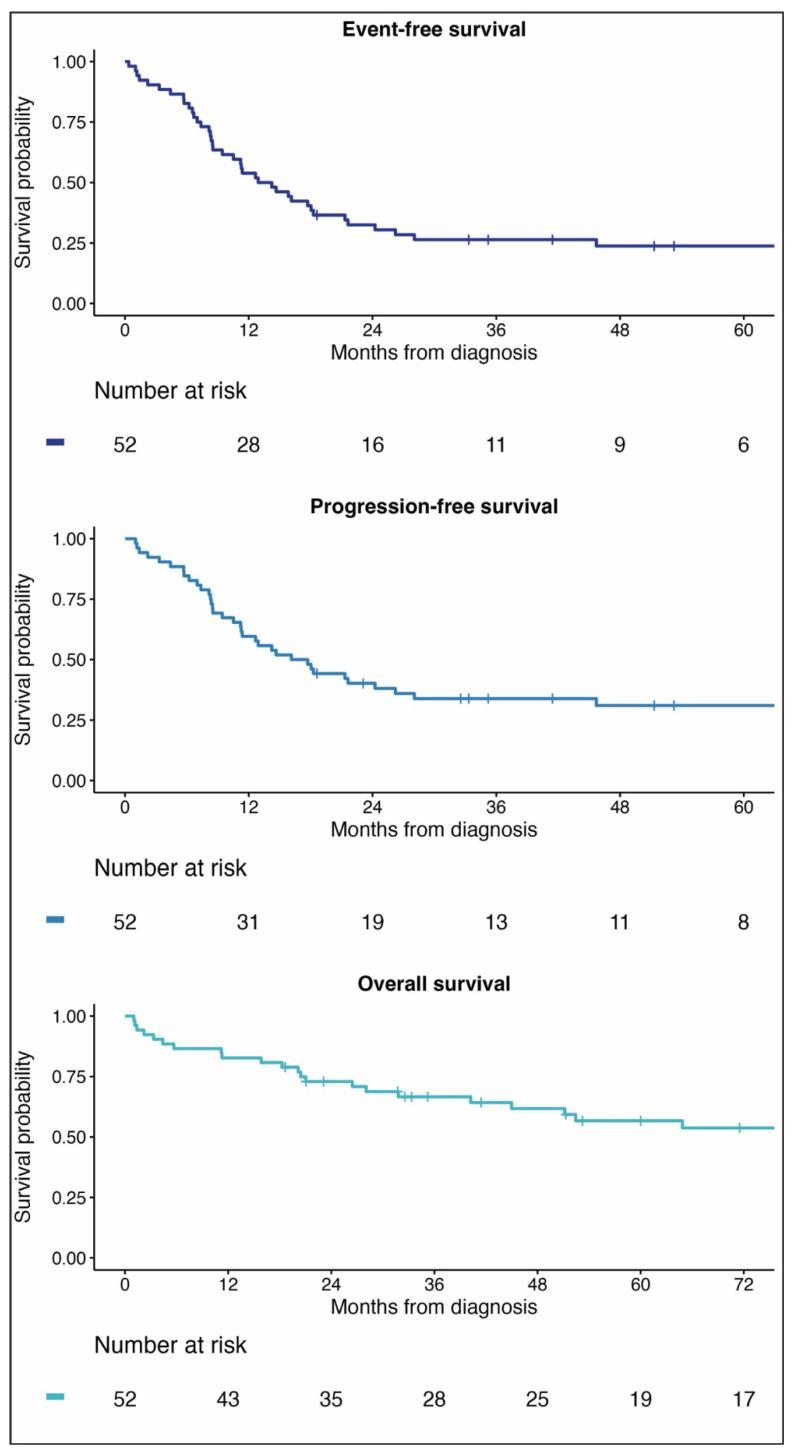
Kaplan–Meier curves of survival of children with Kaposi sarcoma edema.

**Table 1 cancers-16-03769-t001:** Current treatment of Lilongwe Stage 3 and 4 Kaposi sarcoma at Kamuzu Central Hospital.

Kaposi Sarcoma Stage	Preferred Initial Therapy	Alternative Initial Therapy	Limited Response or Relapse
Stage 3	Paclitaxel 100 mg/m^2^ × 6 cycles	Bleomycin 15 units/m^2^ *plus* Vincristine 1.5 mg/m^2^ (BV) × 8 cycles	Paclitaxel cycles as needed *or* Individualized protocol
Stage 4	Paclitaxel 100 mg/m^2^ × 6 cycles	Bleomycin 10 units/m^2^ *plus* Vincristine 1.5 mg/m^2^ *plus* doxorubicin 35 mg/m^2^ (ABV) ^^^	Individualized protocol

All patients with HIV are started on antiretroviral therapy if not already initiated. ^^^ All patients receiving doxorubicin should have a pre-therapy echocardiogram to confirm normal heart function.

**Table 2 cancers-16-03769-t002:** Characteristics of the cohort.

	N = 52 *^a^*
Age (years)	11.7 (9.3, 13.5)
Male sex	33 (63%)
Endemic (HIV-negative) KS	10 (19%)
Location of KS edema	
Lower extremity	47 (90%)
Upper extremity	6 (12%)
Head/neck	5 (9.6%)
Groin/genitals	8 (15%)
Co-occurring sites of KS disease	
Lymph node disease	4 (7.7%)
Skin disease	32 (62%)
Oral disease	10 (19%)
Subcutaneous nodules	13 (25%)
Gastrointestinal or pulmonary disease	9 (17%)
Disseminated skin or oral disease	3 (5.8%)
HAART status at KS diagnosis *^b^*	
HAART-naïve	18 (43%)
Defaulted from previous HAART regimen	1 (2.4%)
On HAART	23 (55%)
CD4 count at KS diagnosis *^b,c^*	
<200 cells/mm^3^	8 (36%)
≥500 cells/mm^3^	6 (27%)
200–500 cells/mm^3^	8 (36%)
HIV viral load at diagnosis *^b,d^*	
Quantitative viral load (copies/mL)	112,896 (275,000–202,503)
HIV viral load undetectable	4 (27%)
Hematologic profile *^e^*	
Platelets <20,000/µL	4 (9.3%)
Hemoglobin <8 g/dL	10 (23%)

Abbreviations: KS, Kaposi sarcoma. *^a^* Median (IQR); n (%); *^b^* HIV-related KS only; *^c^* available for 22 patients; *^d^* available for 15 patients; *^e^* available for 43 patients.

**Table 3 cancers-16-03769-t003:** Most common relapse regiments used during the study period.

Subsequent Line of Therapy	Chemotherapy Regimen	Count
Second-line	Bleomycin–Vincristine	14
Third-line (tie)	Bleomycin–Vincristine	4
Third-line (tie)	Paclitaxel	4
Third-line (tie)	Vincristine only	4
Fourth-line	Paclitaxel	6
Fifth-line	Paclitaxel	4
Sixth-line (tie)	Bleomycin–Vincristine	2
Sixth-line (tie)	Paclitaxel	2
Seventh-line (tie)	Vincristine only	2
Eighth-line (tie)	Paclitaxel	1
Eighth-line (tie)	Vincristine only	1
Eighth-line (tie)	Vincristine + doxorubicin	1
Eighth-line (tie)	Vincristine only	1

**Table 4 cancers-16-03769-t004:** Patterns of relapse among children with Kaposi sarcoma edema.

Initial Stage at Diagnosis	Relapse/Progression Rate		Proportion of Patients Relapsing with Stage 4 Disease	
	HIV-Related KS	Endemic KS	HIV-Related KS	Endemic KS
Lilongwe Stage 3	14/35 (40%)	6/8 (75%)	2/14 (14%)	3/6 (50%)
Lilongwe Stage 4	3/7 (43%)	1/2 (50%)	3/3 (100%)	0/1 (0%)

Abbreviation: KS, Kaposi sarcoma. All patients relapsing as Stage 4 disease relapsed with pulmonary KS.

**Table 5 cancers-16-03769-t005:** Survival of children with Kaposi sarcoma edema at diagnosis by subgroup.

Kaposi Sarcoma Edema Cohort at Diagnosis	Event-Free Survival		Progression-Free Survival		Overall Survival	
	2-Year	5-Year	2-Year	5-Year	2-Year	5-Year
Lilongwe Stage 3 (n = 43)	35% (23–52%)	29% (18–47%)	44% (31–62%)	38% (25–56%)	79% (68–92%)	61% (48–79%)
Lilongwe Stage 4 (n = 9)	22% (7–75%)	0%	22% (7–75%)	0%	44% (21–92%)	33% (13–84%)
HIV-associated KS (n = 42)	33% (22–51%)	25% (15–43%)	43% (30–61%)	34% (22–53%)	71% (59–86%)	57% (43–75%)
Endemic KS (n = 10)	30% (12–77%)	20% (6–69%)	30% (12–77%)	20% (6–69%)	80% (59–100%)	55% (30–100%)

Abbreviation: KS, Kaposi sarcoma. Data are survival % (95% confidence interval).

## Data Availability

The data presented in this study are available on request from the corresponding author. General public access to patient data is restricted by the Malawian National Health Sciences Research Committee.

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
