# Peer review of "Tumor-Associated Edema in Children with Kaposi Sarcoma: 14 Years’ Experience at Kamuzu Central Hospital, Lilongwe, Malawi"

_cancers, 2024, doi:10.3390/cancers16223769_

Round 1
Reviewer 1 Report
Comments and Suggestions for Authors
Congratulations to the authors for the choice of the topic in the manuscript.
the topic is well researched and current.
It would be appropriate to explain why they only included stage 3 and 4 and not stage 1-2
Furthermore, it would be appropriate to indicate in more detail the parameters used to identify recurrence or progression of the disease
Comments on the Quality of English LanguageMinor editing of English language required.
Author Response
Thank you for your thorough review and comments.
RE: including stage 3 and 4 only;
Given its unique phenotype, we included in this study only patients with KS edema (Lilongwe Stage 3 or Lilongwe Stage 4 with KS edema). Except for comparisons between patients with and without KS edema (e.g., comparing age, etc.), we excluded patients with Stage 1 (skin/oral disease only), Stage 2 (lymph node predominant disease) and Stage 4 patients without woody edema. We have clarified this decision-making process in Methods.
Interestingly, 0/141 patients without KS edema at initial presentation went on to develop KS edema with subsequent relapses. We've added this important observation to Results.
For reference, we have provided survival estimates of patients in the larger cohort with Stages 1, 2, and 4 at the bottom of Results (~Line 304).
Parameters to identify recurrence or progression of disease have been added to Methods.
Reviewer 2 Report
Comments and Suggestions for Authors
The authors present a very important analyse of Maposi sarcoma reated in childhood
The improving of some sentences is needed line 197 the data were abstrahed not abstracted. Many failure in description of methodology od study. The references must be better used in text and all references must be in rhe text used. The presentatio of the reault can be imrpoved a tables canbbe bette like the text.
the conclusens can be presented clearly.
the iprovingg is necessar but the presentation is important a can be interesting for many specialist.
The help with languagis failre will by improved the text. This paper may be print in the Cancer.
Author Response
Thank you for your thorough review and comments. We have made the following adjustments based on your review.
- RE: Data were "abstracted". We have changed this to "Data were extracted" for better clarity.
- We have assured that all references are used appropriately in the text.
- The methods and conclusion sections have been clarified.
Reviewer 3 Report
Comments and Suggestions for Authors
Dear authors,
the study is original and analyzes Patients with KS edema with a high risk of relapse and subsequent long-term mortality, even after initial positive treatment responses.
However, the patients were not classified based on visceral disease or additonal infection following the first treatment
could the authors add some information?
Comments on the Quality of English Languageminor spelling errors
Author Response
Thank you for your thorough review and comments.
We have added additional information regarding re-staging of patients with partial response following initial treatment, specifically clarifying patterns of response among the patients with Stage 4 / visceral disease at diagnosis.
Table 4 highlights patients relapsing with visceral disease. A note has been added to the caption indicating that all patients relapsing/progressing as Stage 4 had pulmonary disease. This observation has also been added to the Results.
Round 2
Reviewer 2 Report
Comments and Suggestions for Authors
The disscusions can be imptoved that i am not understood what was the goal of the study. Yours presentation is the report of treatment of HiV positiveor negative cildern with K S.
the teeatment was sed according to treatment of K S
tha survival data are the stanbard results compare the sarcomas and/r K S
what is the reason to wrote it what news you now about the K S childern now.
The readers must be nformed what is the aim of study and what is the nain idea or results new technology new modified therapy …….
Author Response
Thank you for your follow-up review of our resubmission.
We appreciate the reviewers efforts to push us towards clarifying our objectives.
The overarching purpose of this study and much of our group's work is to better characterize the staging/grouping of KS beyond the T-staging system used in the field which poorly captures the heterogeneity of the disease. The study highlights the importance of long-term follow-up of this unique cohort and the need for novel treatment approaches to decrease the risk of relapse and subsequent long-term mortality.
These points have been re-emphasized in the abstract, introduction, methods, and discussion.